# Anterior Quadratus Lumborum Block at the Lateral Supra-Arcuate Ligament versus Transmuscular Quadratus Lumborum Block for Analgesia after Elective Cesarean Section: A Randomized Controlled Trial

**DOI:** 10.3390/jcm11133827

**Published:** 2022-07-01

**Authors:** Min Guo, Bo Lei, Huili Li, Xiaoru Gao, Tianshu Zhang, Ziwei Liang, Yun Wang, Lei Wang

**Affiliations:** 1Department of Anesthesiology, Beijing Haidian Maternal & Child Health Hospital, Beijing 100080, China; minguo973@hotmail.com (M.G.); leibo1977@163.com (B.L.); 18301655633@163.com (X.G.); zhangtianshu17@126.com (T.Z.); lzw732945@163.com (Z.L.); 2Department of Anesthesiology, Beijing Chaoyang Hospital, Capital Medical University, Beijing 100020, China; lily@mail.ccmu.edu.cn

**Keywords:** quadratus lumborum block, lateral supra-arcuate ligament, cesarean delivery, pain, opioid

## Abstract

Several studies have shown the effectiveness of trans-muscular quadratus lumborum block (TQLB) in analgesia after cesarean delivery. However, the influence of anterior QLB at the lateral supra-arcuate ligament (QLB-LSAL) in this surgery is unclear. This study aimed to compare the analgesic efficacy of bilateral TQLBs with bilateral QLBs-LSAL following cesarean delivery. Ninety-four parturients scheduled for cesarean delivery under spinal anesthesia were enrolled and randomly allocated to undergo either bilateral TQLBs or bilateral QLBs-LSAL with 0.375% of ropivacaine (20 mL each side) following cesarean delivery. Intravenous sufentanil was administered for patient-controlled analgesia (PCA). The primary outcome was postoperative sufentanil consumption during the initial 24 h post-surgery. Secondary endpoints included pain scores, time to the first PCA request, postoperative rescue analgesia, satisfaction scores, and nausea/vomiting events. Sufentanil consumption was significantly reduced in the QLB-LSAL group in the first 24 h compared with the TQLB group after surgery (29.4 ± 5.7 μg vs. 39.4 ± 9.6 μg, *p* < 0.001). In comparison with TQLB, the time to the first PCA request in the QLB-LSAL group was significantly longer (10.9 ± 4.1 h vs. 6.7 ± 1.8 h, *p* < 0.001). No differences were observed between two groups regarding pain scores, rescue analgesia after surgery, satisfaction scores, or nausea/vomiting incidence. The significant reduction in opioid consumption in the first 24 h and prolongation in time to first opioid demand in parturients receiving QLB-LSAL compared with TQLB suggest that the QLB-LSAL is a superior choice for multimodal analgesia after cesarean delivery.

## 1. Introduction

Cesarean delivery (CD) is a common obstetric surgical intervention. Pain after CD is associated with the physical pain of the abdominal wall incision and the visceral pain of uterine contraction [1]. Overall, 20% of patients undergoing CD experience moderate to severe pain after surgery [2]. This pain often leads to risk of long-term pain and postpartum depression, which influence postoperative recovery and patient satisfaction, and adversely affect breastfeeding [3,4]. Intravenous patient-controlled analgesia (PCA) has been widely used for postoperative pain management in patients undergoing CD. However, intravenous administrations of opioids by PCA may result in adverse events including pruritis, dizziness, nausea and vomiting, and paralytic ileus. Together with excessive sedation, these adverse events may hinder recovery, aggravate the formation of hypercoagulant thrombosis, and even partly affect the newborn through breastfeeding [5].

Enhanced recovery after surgery (ERAS) protocols have been developed to improve maternal recovery after CD, with one of their important components being the optimization of post-CD analgesia via a multimodal, opioid-sparing strategy [6,7]. Multiple interfascial plane blocks such as the transversus abdominis plane (TAP) block, quadratus lumborum block (QLB), and erector spinae plane block have been widely used for postoperative analgesia after CD [8,9,10,11,12]. In particular, the transmuscular QLB (TQLB) has been demonstrated to be beneficial for CD [12,13]. TQLB was first described by Borglum in 2013 [14] and involves the injection of local anesthetic (LA) within the interfascial space between the QL and psoas major muscles at the L3/4 level to result in the lower thoracic and lumbar paravertebral blocks.

Recently, our group has developed a novel anterior QLB described as QLB at the lateral supra-arcuate ligament (QLB-LSAL), which is characterized by rapid onset time, wide dermatomal coverage of sensory block (T6–7–L1–2), and a low incidence of muscle weakness in lower limbs [15,16,17]. With QLB-LSAL, LA is directly injected into the compartment between the diaphragm and QL muscle, bypassing the lateral arcuate ligament barrier, which facilitates LA diffusion into lower thoracic paravertebral space and thereby achieves wider somatic/visceral coverage. However, the utility of anterior QLB-LSAL as a component of multimodal postoperative analgesia for CD patients remains to be established. Therefore, in this study, we aimed to compare the efficacy of bilateral anterior QLBs-LSAL with that of bilateral TQLBs in reducing postoperative opioid consumption following CD.

## 2. Methods

### 2.1. Study Design and Setting

Ethical approval for this study was obtained from the Ethics Committee of Beijing Haidian Maternal & Child Health Hospital, China, on 12 May 2020 (reference number: 2018-26-01-2020). The study was registered with the Chinese Clinical Trial Registry (ChiCTR2100043063) on 4 February 2021, and was performed in line with the Consolidated Standards of Reporting Trials (CONSORT) statement and the Declaration of Helsinki (Figure 1).

### 2.2. Subjects

This single-center, randomized, and controlled study was conducted at Beijing Haidian Maternal & Child Health Hospital. After acquiring written informed consent, 102 parturients were designated to receive elective CD under spinal combined with epidural block and were enrolled between 1 March 2021 and 12 September 2021 (Figure 1). Eligibility requirements for inclusion in this study were an age between 18 and 40 years, a normal singleton pregnancy with a gestation of at least 36 weeks, a body mass index (BMI) less than 30 kg/m^2^, and an American Society of Anesthesiologists (ASA) physical status of 1–2. Parturients were excluded if they declined to participate in the study, had local or systemic infections or blood coagulation disorders, were allergic to LAs or opioids, suffered from alcohol abuse, had a history of previous lumbar spinal surgery, suffered from cognitive dysfunction before the operation and/or unable to cooperate with pain scoring or to use the PCA system, or difficulties were experienced in visualizing muscular and fascial structures on the sonogram. Basic demographics including age, height, weight, ASA physical status classification, and BMI were recorded.

### 2.3. Anesthesia Management

After entering the operating room, patients were monitored with three-lead electrocardiography, pulse oxygen saturation, and noninvasive blood pressure monitoring. Peripheral vein access was then obtained, and intravenous infusion using Ringer’s lactate was initiated. A standardized regimen for spinal combined with epidural block was administered at the L2~3 or L3~4 intervertebral spaces at the lateral decubitus position. The spinal solution consisting of 1 mL of ropivacaine (1%, Batch no. NAWG, AstraZeneca AB, Sodertalje, Sweden) and 1 mL of cerebrospinal fluid was administered at an infusion rate of 1 mL/6 s. The epidural catheter was inserted for stand-by use.

### 2.4. Study Intervention

Participants were designated to receive either bilateral TQLBs or bilateral QLBs-LSAL using a computer-randomized list with a 1:1 intergroup ratio. A nurse anesthetist who was blinded to the blocks set up 2 × 20 mL syringes containing saline to ensure the correct plane, together with 2 × 20 mL syringes containing 0.375% ropivacaine, marked with the randomized identification number for the individual patient. The overall ropivacaine dose limit for each patient was 40 mL. The investigators involved in the outcomes assessment and the participants were blinded to the group designations.

### 2.5. Surgery and Block Procedure

All elective CD procedures were performed using the Joel-Cohen incision as routine practice. The Joel-Cohen incision is placed 3 cm caudad to the anterior intercristal line. Following the surgery, the patients were transferred to the postanesthesia care unit (PACU) and monitored, after which the bilateral TQLBs or QLBs-LSAL was performed under ultrasound guidance with a convex probe (5~8 MHz, Sonosite M-Turbo Portable Ultrasound System, SonoSite, Bothell, Washington, DC, USA). All blocks were performed by the anesthesiologists from our Acute Pain and Regional Anesthesiology Service (MG, BL, and LW). They have more than 10 years of experience in regional anesthesia and acute pain management, and each was experienced in ultrasound-guided TQLB or QLB-LSAL (>50  blocks).

To perform the TQLBs, the patients were placed in the lateral position. After sterile preparation, the transducer was placed above the iliac crest at the posterior axillary line for conducting a transverse scan. The shamrock sign was seen on the sonogram [18]. The puncture needle (22-gauge, 100 mm, Tuoren™, Henan, China) was inserted from the posterior side of the ultrasound probe in the ultrasound beam plane in a posterolateral-to-medial direction until the needle tip pierced the QL muscle to the interfascial plane between the QL and psoas major muscles [10,12,19]. The needle endpoint was then confirmed by the injection of 2–5 mL of 0.9% sodium chloride, and ropivacaine (20 mL; 0.375%) was slowly introduced within target fascial-interspace posterior to the transversalis fascia (Figure 2). The TQLB was then performed on the contralateral side.

To perform the QLBs-LSAL, patients were placed in the lateral position. After cleaning the skin, the transducer was placed above the L1 transverse process (TP) tip and T12-rib to perform a parasagittal scan. Thus, the apposition zone between the QL and diaphragm could be identified on the sonogram [17,20]. The needle was inserted in-plane from the caudal edge of the transducer until the needle tip pierced the QL muscle fiber and accessed the apposition zone between the diaphragm and the QL muscle. The ropivacaine (20 mL; 0.375%) was then gradually injected following multiple negative aspirations (Figure 3). QLB-LSAL was then conducted on the contralateral side.

### 2.6. Postoperative Pain Management and Outcomes Assessments

After the bilateral blocks, both groups of patients received intravenous sufentanil administration via a PCA pump (TR-10-100 pump; Tuoren, Henan, China), which was formulated as sufentanil (100 μg; Eurocept BV, Ankeveen, The Netherlands) and tropisetron (5 mg; Lingkang Pharma, Hainan, China) diluted to 100 mL. The final concentration of sufentanil in the infusion was 1 μg/mL. The pump was programmed to deliver a 2 mL intravenous bolus on demand, with a lockout interval of 15 min and 1 mL/h background infusion. Participants were educated on the number-based ranking scale (NRS)—where 0 was equivalent to ‘no pain’ and 10 was ‘worst pain imaginable’—and use of the PCA pump. Two anesthetists blinded to the group allocations assessed the NRS pain scores at rest and on movement for each patient at 4, 8, 12, and 24 h postsurgery. A single cough was required for assessing the pain score on movement. If the patient’s NRS still exceeded 4/10 at rest with the PCA bolus, intravenous Tramadol^®^ (100 mg; Grunenthal™, Stolberg, Germany) was injected for rescue analgesia (0.1 mg morphine being equivalent to 1 mg Tramadol^®^).

The total opioid consumption in the first 24 h postsurgery was recorded as the primary outcome. Secondary endpoints included the NRS pain scores, satisfaction scores (1 = ‘very satisfied’, 2 = ‘satisfied’, 3 = ‘dissatisfied’, and 4 = ‘very dissatisfied’), and postsurgical adverse events during the initial 24 h postsurgery, including postoperative nausea and vomiting (PONV), pruritis, hypotension, shivering, and respiratory depression (<10 breaths/min). The quadriceps femoris weakness at 8 h after surgery was also assessed [21].

### 2.7. Sample Size Calculation and Statistical Analysis

The sample size of the study was chosen according to prior observations at our institution that an intravenous sufentanil dosage (mean ± SD) of 40.0 ± 11.8 μg was needed in the first 24 h postsurgery in parturients who received postoperative TQLB. Assuming that QLB-LSAL would decrease the required sufentanil dosage by 25% relative to the TQLB group, an estimated 84 patients (42/group) were deemed necessary for the present study when the study power was set to 0.8 (two sides). This power calculation was performed using an online tool (https://www.cnstat.org/samplesize/, accessed on 20 December 2020), with a type I error associated with a test for the null hypothesis of 0.05. Considering the possible dropout of up to 10% of participating patients, a target sample size of 92 patients (46/group) was planned.

A standardized protocol form was employed to collect all raw data. All numerical and categorical data were stored on a computer. SPSS^®^ (V.21.0; IBM Corp., Armonk, NY, USA) was used for statistical analysis. The Kolmogorov–Smirnov test was employed for evaluating the distribution normalities of variables. Categorical data expressed as percentages were compared with chi-square (χ^2^) or Fisher’s exact tests. Quantitative data are expressed as mean ± SD or median [IQR]. Independent *t*-tests or Mann–Whitney U tests were conducted for assessing the differences between the two groups. Repeated-measures analysis of variance was performed within groups for variables with normal distribution. Log-rank tests were used in conjunction with Kaplan–Meier plots for the duration of time until the first opioid request. *p* < 0.05 was considered to be statistically significant.

## 3. Results

One hundred and two patients were enrolled in the study (Figure 1). Five patients refused to participate in the study. Three patients were excluded due to poor ultrasound imaging quality. The remaining 94 patients were randomly assigned into two groups (*n* = 47/group). One patient in each group experienced postoperative hemorrhage and was withdrawn from the study. The data from 46 patients in each group were finally analyzed. There were no significant differences in demographic data (age, ASA physical status classifications, BMI, or surgical duration time) between the two groups (all *p* > 0.05; Table 1). No parturients were administered LA epidural infusions.

Patients that had received bilateral QLBs-LSAL exhibited a significant 26.4% reduction in cumulative sufentanil use during the first 24 h postsurgery in comparison with patients in the TQLB group (29.4 ± 5.7 μg vs. 39.4 ± 9.6 μg, *p* < 0.001; Table 2). The time to first PCA request in the QLB-LSAL group was markedly prolonged in comparison with TQLB (10.9 ± 4.1 h vs. 6.7 ± 1.8 h, *p* < 0.001; Figure 4).

No differences in the reported pain intensity at rest and on movement between the two groups were observed at any of the follow-up time-points (*p* > 0.05; Table 2). The opioid-linked adverse-event incidence (nausea, emesis, and pruritus) was comparable between the two groups (*p* > 0.05; Table 2). The satisfaction scores did not differ between the groups (*p* > 0.05; Table 2).

There was no significant difference in the number of patients that required rescue analgesia between the two groups (*p* > 0.05; Table 2). One patient in the TQLB group had lower limb weakness, while no patients in either group suffered from systemic LA toxicity or respiratory depression (*p* > 0.05; Table 2).

## 4. Discussion

This study showed that bilateral QLB-LSA resulted in a 26.4% decrease in the use of sufentanil during the first 24 h after surgery in comparison with TQLBs in patients undergoing CD. In addition, a significant prolongation of time to the first opioid request was observed. 

The pain experienced after CD is complex and consists of somatic pain originating from the abdominal wall incision located in the cutaneous segment area innervated by the anterior branch of the T12~L1 spinal nerves and the rectus abdominis dissected in the surgical area innervated by the anterior branches of the T6~T7 thoracic nerves [22]. The pain also has a visceral component due to uterine exteriorization and stretching innervated by the T5~T10 thoracic sympathetic nerve [23]. After surgery, breastfeeding increases oxytocin release to stimulate uterine contraction. The application of uterine contraction stimulators and pressing the fundus also increase postoperative pain. Therefore, nerve blocks are used to obtain better postoperative analgesia in parturients after CD.

The TQLB procedure deposits the LA within the interfascial space between the QL and psoas major muscles just deep to the transversalis fascia and below the medial and lateral arcuate ligaments, allowing cranial diffusion of the injectate to the lower thoracic paravertebral space [24]. As shown by both cadaveric and clinical studies, the TQLB can achieve sensory coverage of the T10~L4 dermatomes and is assumed to block the sympathetic nerves and thus reduce visceral pain [12,25,26]. The TQLB has been reported as an approach to reduce postoperative opioid use, prolonging the time for first opioid demand in CD cases [12,23,27,28]. However, the procedure has several shortcomings, including its relatively slow onset time, excessive deposition of LA in the lumbar fascia space, and the dependence on the integrity of the transversalis fascia [29], which limits its use to some extent. To enhance analgesic effectiveness, our team created a QLB-LSAL strategy where LA is directly inserted anterior to the QL muscle at the LSAL, consequently avoiding the LSAL barrier and, therefore allowing quicker LA diffusion into the lower thoracic paravertebral space, ultimately achieving a thoracic paravertebral block linked to the dermatomal coverage of T6–7 to-L1–2 [4,14]. 

Our previous study demonstrated that QLB-LSAL was associated with a 31.5% reduction in sufentanil use during the first 24 h postsurgery in comparison with the TQLB approach after laparoscopic nephrectomy [16]. However, it remains unclear whether QLB-LSAL is beneficial for postoperative analgesia in parturients after CD. In the current study, a 26.4% decrease in sufentanil consumption was observed during the initial 24 h post-CD in the QLB-LSAL group compared with the TQLB group, suggesting the higher analgesic efficacy of QLB-LSAL. The intravenous sufentanil consumption in the 24 h postoperative period in the TQLB group was 39.4 ± 9.6 μg. In Hansen’s study, the first 24 h oral morphine consumption in parturients was about 65.3 mg in the TQLB group, which was lower than that in our study [12]. However, Hansen et al. administered 30 mL of 0.375% ropivacaine on each side and all patients regularly received oral paracetamol and ibuprofen, which may explain the difference. The optimum volume and concentration of ropivacaine used with the TQLB need to be further investigated.

Our previous study showed that the QLB-LSAL can contribute sensory-block coverage of T6–7–L1–2 [15,17]. Wider block segments may partly explain the significant opioid-sparing action of QLB-LSAL relative to the TQLB in parturients undergoing CD. Additionally, considering the limitation of the maximum dose of LA in bilateral blocks, we used a lower concentration and volume of LA (0.375% ropivacaine, 20 mL/side). This may have resulted in the difference in the reduced sufentanil consumption in the QLB-LSAL group relative to the TQLB group between laparoscopic nephrectomy and CD. 

The NRS pain scorings at 4, 8, 12, and 24 h after elective CD at rest and on movement in patients receiving bilateral TQLB and QLB-LSAL in our study were relatively low, and no significant difference between groups was observed. However, the time to first PCA request in the QLB-LSAL group was markedly longer in comparison with the TQLB group (10.9 ± 4.1 h vs. 6.7 ± 1.8 h). In Hansen’s study, the time to first opioid request in the TQLB cohort was 5.6 h, which was similar to that in our current study [12]. A retrospective study showed that while postoperative analgesia with QLB compared with intrathecal morphine for elective CD was associated with an increase in the time to first opioid use, it did not decrease the pain score [30]. 

No hypotension, shivering, or respiratory depression (<10 breaths per minute) was experienced by any of the patients. Five patients in the TQLB group received Tramadol for rescue analgesia 6–10 h after surgery, while four patients in the QLB-LSAL group received Tramadol for rescue analgesia 8–12 h post-surgery. There were no statistically significant differences between the groups regarding the incidence of other secondary outcomes, including patient satisfaction scores or opioid-linked adverse events. No patients developed pneumothorax during the study. There is evidence that TQLB affects the lumbar plexus causing quadriceps weakness in patients undergoing hip surgery [31]. In the current study, only one patient in the TQLB group experienced lower limb weakness despite the use of lower LA concentrations. Lower limb weakness may prolong the time of the first “walk out of the bed” and increase the risk of venous thrombosis in the lower extremities.

Our study has some limitations. First, the relatively small sample size could restrict the recognition of rare adverse events using this new block. Secondly, we did not test the block segments of QLB-LSAL or TQLB, because the blocks were performed immediately after surgery and the spinal anesthesia was still maintained. Additionally, evaluating block success (dermatomes and myotomes affected) had the risk of unblinding the patient, staff, or investigators to block allocation. Third, we did not administer intrathecal opioids after CD to avoid interfering with the interfascial plane blocks. The issue of whether the addition of QLB to parturients receiving neuraxial morphine results in additional analgesic benefit compared with neuraxial morphine alone is still a point of discussion in meta-analyses and clinical studies [22,32,33]. Fourth, all parturients in the current study received the Joel-Cohen incision, but not Pfannenstiel incision. The different approaches of QLBs for analgesia in patients with Pfannenstiel incision deserve further investigation.

## 5. Conclusions

This study showed that as an interfascial plane block, the QLB-LSAL approach may be more beneficial for postoperative analgesia in parturients undergoing CD than TQLB because it significantly reduces cumulative sufentanil consumption in the 24 h following surgery, thus prolonging the time to first analgesic request. 

## Figures and Tables

**Figure 1 jcm-11-03827-f001:**
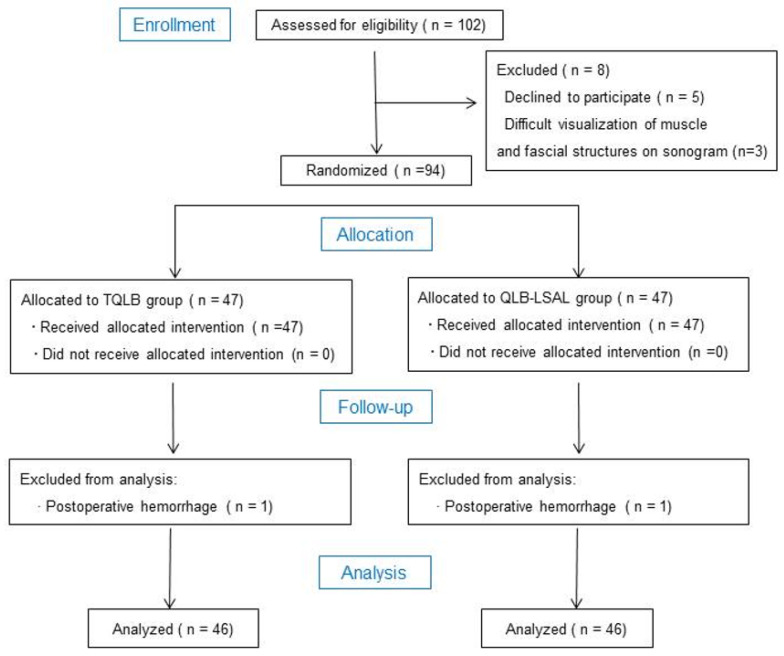
CONSORT study design. CONSORT, Consolidated Standards of Reporting Trials; TQLB, transmuscular quadratus lumborum block; QLB-LSAL, quadratus lumborum block at the lateral supra-arcuate ligament.

**Figure 2 jcm-11-03827-f002:**
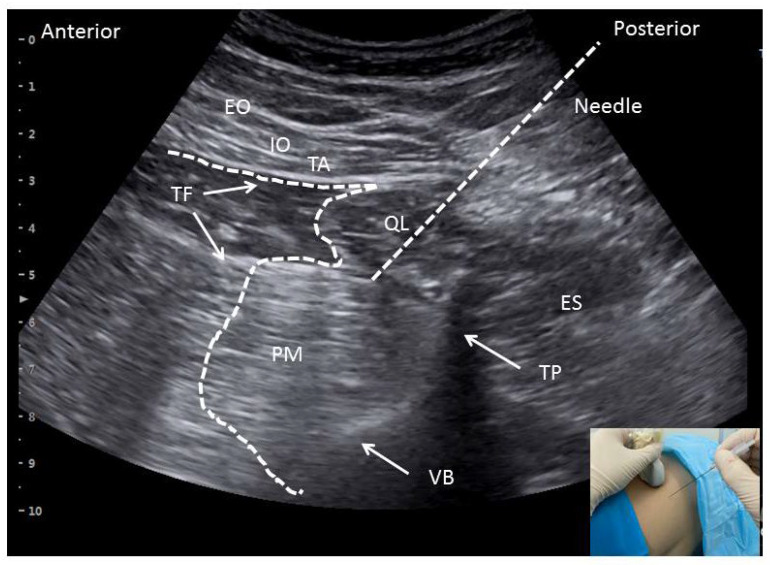
TQLB block under the shamrock sign on the sonogram. TQLB, transmuscular quadratus lumborum block; QL, quadratus lumborum; ES, erector spinae; PM, psoas major; VB, vertebral body; TP, transverse process; TF, transversalis fascia; EO, external oblique abdominis; IO, internal oblique abdominis; TA, transverse abdominis.

**Figure 3 jcm-11-03827-f003:**
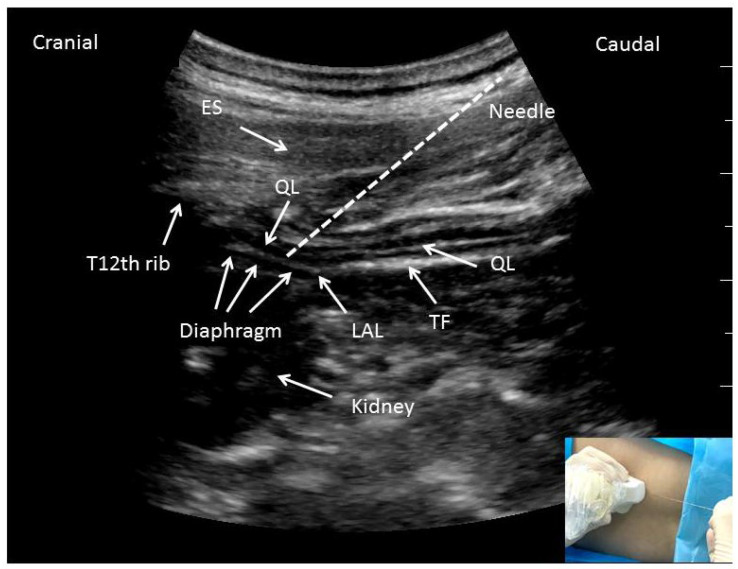
Parasagittal anterior QLB-LSAL under ultrasound guidance. QLB-LSAL, quadratus lumborum block at the lateral supra-arcuate ligament; QL, quadratus lumborum; ES, erector spinae; LAL, lateral arcuate ligament; TF, tranversalis fascia.

**Figure 4 jcm-11-03827-f004:**
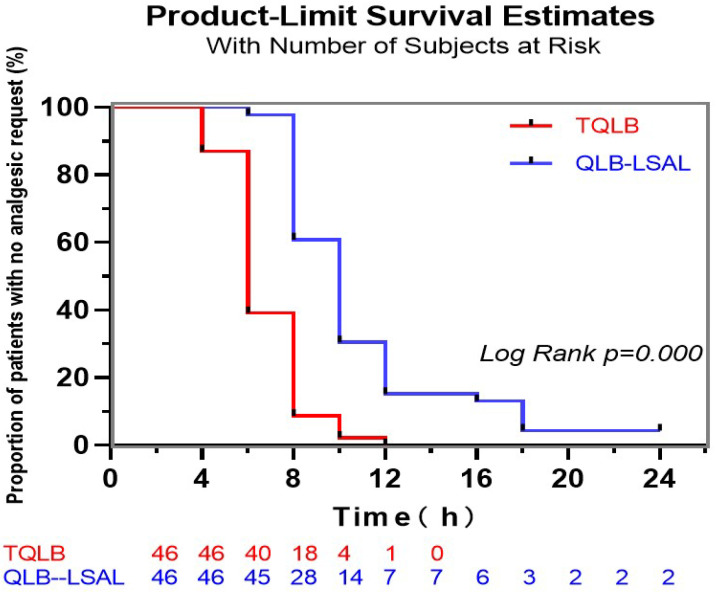
Kaplan–Meier estimate of time to first opioid administration. The participant’s ‘survival’ ended with the first opioid administration.

**Table 1 jcm-11-03827-t001:** Demographic and clinical characteristics.

Variable	TQLB Group	QLB-LSAL Group
Number	46	46
Age (years)	31.7 ± 4.6	32.7 ± 3.6
Height (cm)	160.3 ± 5.4	159.8 ± 4.8
Weight (kg)	70.1 ± 8.8	70.5 ± 8.0
BMI (kg/m^2^)	27.4 ± 6.5	27.9 ± 4.6
ASA classification, *n* (%)		
ASA class I–II	43 (93.5%)	42 (91.3%)
ASA class III	3 (6.5%)	4 (8.7%)
Duration of surgery (min)	41.5 ± 8.3	43.2 ± 6.8
Blood loss (mL)	320.6 ± 42.5	312.8 ± 36.4

Numerical variables are expressed as mean (SD). Categorical variables are expressed as number (percentage). BMI, body mass index; ASA, American Society of Anesthesiologists.

**Table 2 jcm-11-03827-t002:** The postoperative data.

	TQLB Group	QLB-LSAL Group	*p*
Intravenous sufentanil consumption in first 24 h after surgery (μg)	39.4 ± 9.6	29.4 ± 5.7	*p* < 0.001
Number of cases requiring rescue analgesia during initial 24 h (%)	5 (10.8%)	4 (8.7%)	0.727
Postoperative pain intensity at rest, median (IQR)			
NRS at 4 h	2.0 (1.0–3.3)	1.8 (1.2–2.4)	0.191
NRS at 8 h	2.2(1.2–3.1)	2.1 (1.4–2.8)	0.761
NRS at 12 h	2.3 (1.6–3.0)	1.8 (1.2–2.7)	0.064
NRS at 24 h	2.3 (1.5–3.7)	2.2 (1.4–3.2)	0.305
Postoperative pain intensity on movement, median (IQR)			
NRS at 4 h	2.9 (1.5–4.9)	2.6 (1.9–3.5)	0.417
NRS at 8 h	3.2 (2.6–4.2)	2.9 (2.3–4.4)	0.400
NRS at 12 h	3.5 (2.7–4.4)	3.2 (2.5–4.4)	0.351
NRS at 24 h	3.7 (2.5–5.2)	3.6 (2.9–4.5)	0.635
First time to opioid request (h)	6.7 ± 1.8	10.9 ± 4.1	*p* < 0.001
Postoperative nausea	2 (4.3%)	3 (6.5%)	0.647
Episodes of vomiting	2 (4.3%)	2 (4.3%)	1.000
Pruritus	2 (4.3%)	2 (4.3%)	1.000
lower limb weakness	1 (2.2%)	0 (0.0%)	0.317
Patient satisfaction score, median (IQR)	1 (1–2)	1 (1–2)	0.525

Values are median [IQR] or number of patients (%). NRS, numerical rating scale.

## Data Availability

The datasets analyzed during the current study are not publicly available due to ethical reasons but are available from the corresponding author on reasonable request.

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
