# Peer review of "Anterior Quadratus Lumborum Block at the Lateral Supra-Arcuate Ligament versus Transmuscular Quadratus Lumborum Block for Analgesia after Elective Cesarean Section: A Randomized Controlled Trial"

_jcm, 2022, doi:10.3390/jcm11133827_

Round 1

Reviewer 1 Report

The main question addressed by the research is postoperative pain relief after caesarean section. It is relevant and interesting. The topic is not original but has interest for adding the use of a type of neural blockade that is not usually used in this type of surgery. Compared with other published material this paper add information about using Transmuscular Quadratus Lumborum block for analgesia after cesarean section and it may be more beneficial for postoperative analgesia after cesarean section . The paper is well written and the text isclear and easy to read. The conclusions are consistent with the evidence and arguments presented and they address the main question posed.

Author Response

Response to the Reviewers’ Comments

Dear Editors and Reviewers:

Thank you very much for giving us an opportunity to revise our manuscript. We appreciate for Editors/Reviewers’ warm work earnestly, and these valuable comments helped us thoroughly improving the manuscript. The manuscript has been revised in accordance with these proposals. The revised portion is marked in red in the manuscript and responses to your comments are as follows:

Response to the Reviewer #1:

Comment: The main question addressed by the research is postoperative pain relief after caesarean section. It is relevant and interesting. The topic is not original but has interest for adding the use of a type of neural blockade that is not usually used in this type of surgery. Compared with other published material this paper adds information about using Transmuscular Quadratus Lumborum block for analgesia after cesarean section and it may be more beneficial for postoperative analgesia after cesarean section. The paper is well written and the text is clear and easy to read. The conclusions are consistent with the evidence and arguments presented and they address the main question posed.

Response: We thank the Reviewer#1 for the beneficial comments.

Reviewer 2 Report

I read with great interest the manuscript titled "Anterior Quadratus Lumborum Block at the Lateral Supra-arcuate Ligament versus Transmuscular Quadratus Lumborum Block for Analgesia after Elective Cesarean Section: A Randomized Controlled Trial" (ID jcm-1747742).    

In my honest opinion, the topic is fascinating enough to attract the readers' attention. The methodology is accurate, and the data analysis supports conclusions. Nevertheless, the authors should clarify only two points:

1) LINE 42: The authors did not cite a novel key article about the topic: https://doi.org/10.1002/ijgo.13563.

2) The authors have not appropriately highlighted a fourth study's limitations:  Only a Joel-Cohen incision was utilized for the caesarean section (line 111). The caesarean section is typically conducted through a Pfannenstiel incision, which may have a minimal impact on postoperative pain. These points should be better defined, in my opinion.

Although the manuscript is well-written, minor issues need to be addressed before further consideration.

Reviewer 3 Report

Article explained in an interesting and novel way for the treatment of post-cesarean section pain. when the article is examined;

1.     According to the recommendations of ESRA prospect (2020), ‘If IT morphine is not used, quadratus lumborum block is recommended (Grade A) for its effect in reducing pain scores and opioid requirements.’ But QLB –LSAL tecnique should be questioned for side effects TQLB is already effective between T6-7 and L1-2 dermatomes. this technique safe and easy

2.     It is necessary to conduct cadaveric studies related to the QLB-LSAL technique described in this study.

3.     While the classical QLB method can be applied and a sufficient sensory level can be achieved, I do not find it appropriate to apply this technique, which may have many complications.

4.     The material and the method of the study are not well established.

5.     Unfortunatelly sensory level was not checked after the block. This is  great shortcoming for this study

6.     In my opinion, tramadol, which is used for rescue analgesia, is not suitable because it is a weak opioid. According to the recommendations of ESRA prospect (2020), nonsteroidal anti-inflammatory drug should be added in the treatment of post-cesarean section pain

7.     In addition, the authors mentioned to the patients that they had a spino-epidural block.

8.     But epidural analgesia was not mentioned in the treatment of postoperative pain. The operation was managed with spinal anesthesia. So where and when was epidural anesthesia/analgesia used?.

 Article explained in an interesting and novel way for the treatment of post-cesarean section pain. when the article is examined;

1.     According to the recommendations of ESRA prospect (2020), ‘If IT morphine is not used, quadratus lumborum block is recommended (Grade A) for its effect in reducing pain scores and opioid requirements.’ But QLB –LSAL tecnique should be questioned for side effects TQLB is already effective between T6-7 and L1-2 dermatomes. this technique safe and easy

2.     It is necessary to conduct cadaveric studies related to the QLB-LSAL technique described in this study.

3.     While the classical QLB method can be applied and a sufficient sensory level can be achieved, I do not find it appropriate to apply this technique, which may have many complications.

4.     The material and the method of the study are not well established.

5.     Unfortunatelly sensory level was not checked after the block. This is  great shortcoming for this study

6.     In my opinion, tramadol, which is used for rescue analgesia, is not suitable because it is a weak opioid. According to the recommendations of ESRA prospect (2020), nonsteroidal anti-inflammatory drug should be added in the treatment of post-cesarean section pain

7.     In addition, the authors mentioned to the patients that they had a spino-epidural block.

8.     But epidural analgesia was not mentioned in the treatment of postoperative pain. The operation was managed with spinal anesthesia. So where and when was epidural anesthesia/analgesia used?.
